# Prevalence of anxiety and depression symptoms and associated factors among women diagnosed with breast cancer: A cross-sectional study at Ocean Road Cancer Institute in Dar es Salaam–Tanzania

**Ummy Msenga¤*, Joel S. Ambikile, Salome E. Buluba**

Department of Clinical Nursing, School of Nursing, Muhimbili University of Health and Allied Sciences, Dar es Salaam, Tanzania

¤ Current address: Department of Mental Health and Psychiatric Nursing, School of Nursing, Kairuki University, Dar es Salaam, Tanzania.
* msengaummy@gmail.com

## Abstract

### Introduction

Anxiety and depression are the prevailing mental illnesses in low and middle-income nations. The shock of receiving breast cancer diagnosis and it's effects on daily life make it perplexing to adjust to the situation, hence anxiety and or depression symptoms develop. This study aimed to determine the prevalence and factors associated with anxiety and depression symptoms among women diagnosed with breast cancer at Ocean Road Cancer Institute (ORCI) in Dar es Salaam, Tanzania.

### Methods

We conducted an analytical quantitative cross-sectional study between May and June 2023 among 384 women diagnosed with breast cancer using consecutive sampling. We used a pre-tested interviewer-administered questionnaire to collect data and the Statistical Package for Social Sciences (IBM SPSS) version 23 for data analysis. Bivariate and multivariate logistic regression analyses were performed to identify factors associated with anxiety and depression symptoms, and a p-value <0.05 was considered statistically significant.

### Results

We found a relatively high prevalence of anxiety (44.8%) and depression (50.5%). Participants who didn't know their stage of cancer [adjusted Odds Ratio (aOR): 0.39; 95% Confidence Interval (CI): 0.19, 0.83; p = 0.014]; in early stage of cancer [aOR: 0.36; 95%CI: 0.16, 0.81; p = 0.014]; with low state of hope [aOR: 0.52; 95%

**Data availability statement:** Due to participant privacy concerns and restrictions imposed by the ethics approval granted by the National Institute of Medical Research, the authors are unable to publicly share the full dataset for this study. Researchers interested in accessing the data for academic purposes may request access and such requests will be reviewed to ensure compliance with ethical and legal obligations. The contact details for NatREC (National Health Research Ethics Sub-Committee) in Tanzania is as follows: Telephone: +255 22 212 14 00, Physical address: 2448, Ocean Road P.O.Box 9653 Dar es Salaam Tanzania and Email: ethics@nimr.or.tz.

**Funding:** The author(s) received no specific funding for this work.

**Competing interests:** The authors do not have any competing interest.

CI: 0.31, 0.86; p = 0.012]; and low cognitive reappraisal [aOR: 2.09; 95%CI: 1.33, 3.28; p < 0.001] were significantly associated with anxiety. Those without partners [aOR: 2.56; 95%CI: 1.59, 4.10; p < 0.001]; didn't know their stage of cancer [aOR: 0.38; 95%CI: 0.17, 0.86; p = 0.020]; receiving radiotherapy treatment [aOR: 3.86; 95% CI: 1.34, 11.08; p = 0.012]; and low cognitive reappraisal [aOR: 2.41; 95%CI: 1.52, 3.83; p < 0.001] were significantly associated with depression.

## Conclusion

Symptoms of depression and anxiety were very common among women diagnosed with breast cancer in this study. We strongly advise for proper care including routine screening from mental health specialists to improve treatment outcomes among this population group.

## Introduction

Globally, breast cancer ranks as the second most common cause of death among women [1]. Since breast cancer is a chronic and terminal illness, it can be difficult to cope emotionally resulting into anxiety and depression following a cancer diagnosis [2–8]. Even though the success rate of breast cancer treatment has increased dramatically over the years due to advancement in treatment modalities like radiation and chemotherapy, survivors still experience unsettling levels of psychological burden [4,8].

The Diagnostic and Statistical Manual of Mental Disorders, Fifth Edition Text Revision (DSM-5-TR) defines anxiety as excessive worry and apprehensive expectations about a variety of events or activities, such as work, getting ill or school performance, that occur more often than not for at least six months. In contrast depression is defined as a mood disorder that lasts for at least two weeks and is characterized by a persistent sense of sadness and feelings of loss of interest in once enjoyed activities. The physical symptoms must result in clinically substantial distress or hinder social, occupational, or other critical areas of functioning in order to be diagnosed as either depression or anxiety [9].

After receiving a breast cancer diagnosis and therapy, globally between 25% and 33% of individuals experience discomfort, anxiety, and depression [3,4]. The prevalence globally in 2020, Africa in 2016 and Tanzania in 2022 of anxiety among breast cancer patients was 18%, 19%, and 9.9%, while that of depression was 21%, 25%, and 20.6%, respectively [5–7].

The shock of receiving a breast cancer diagnosis, ongoing family issues, financial responsibilities, and social obligations, the financial strain of treatment, changes in one's body, and one's self-image provide significant obstacles adjusting to and coping with the circumstances. The majority of patients experience emotional distress that qualifies as a diagnosable mental illness [10,11]. Anxiety and depression increase the likelihood of poor drug adherence and unhealthy lifestyles. They can

also worsen the course of breast cancer treatment and limit an individual's capacity to work, which increases financial burden, lowers quality of life, and raises the risk of dying young [5,10,11].

Previous studies have identified several factors associated with anxiety and depression symptoms among women diagnosed with breast cancer.Most studies found that early stage of cancer, marital status that is being without a partner and change in body image due to the disease process and treatment side effects were associated with anxiety and depression symptoms at different levels [5,6].

In Tanzania it has been reported that the patient's socioeconomic situation had more to do with the anxiety and depression than the actual pathology or therapy of her breast cancer. A couple of things like not having enough money and going through a heartbreaking recent event like a loved one's death were associated [7]. However, research on the psychological challenges among Tanzanian women diagnosed with breast cancer is limited. Therefore, this study aimed to determine the prevalence and factors associated with anxiety and depression symptoms among women diagnosed with breast cancer at Ocean Road Cancer Institute (ORCI) in Dar es Salaam, Tanzania.

## Methodology

### Study design and setting

This was an analytical cross-sectional study that employed a quantitative approach. This study was conducted at ORCI, Dar es Salaam region in Tanzania mainland. ORCI is located in Ilala district, Chimara Street and it is the only cancer institute in the country. It has a bed capacity of 270 patients and provides outpatient services to 200–250 patients per day, Approximately, 500–600 breast cancer patients are attended at the outpatient department monthly. The institute offers a variety of therapies for cancer patients, including chemotherapy, radiotherapy, brachytherapy, and palliative care [12]. ORCI was chosen purposively considering the nature of participants needed for this study.

### Study population and selection criteria

This study involved women aged 18 years and above, diagnosed with breast cancer for more than six months, and attending outpatient clinics during the study period. Those who consented to participate in the study were included. However, those who could not understand Swahili and English, very sick, nauseous and unable to make a conversation due to medication side effects were excluded.

### Sample size and sampling procedure

We determined the sample size using Cochran's sample size calculation formula as follows:

$$N = Z^2 P (100 - P)/e^2$$

Where;
N = the required minimum sample size
e = margin error (5%)
P = prevalence of women with anxiety and depressive symptoms after being diagnosed with breast cancer which is 50% [13].
Z = standard normal deviate corresponding to 95% confidence level (1.96)

$$= (1.96)^2 \, 50 \, (100 - 50)/5^2$$
$$= 9604/25$$
$$N = 384$$

The sample size was 384 participants.

We recruited participants using consecutive sampling, a technique deemed suitable for this study due to the nature of the study participants. The participants attendance to the hospital are affected by their financial situation, family/significant other support and or planned follow-up clinics. We approached all adult-aged women diagnosed with breast cancer attending their scheduled follow-up and treatment clinics like chemotherapy and radiotherapy, from 4th May to 17th June 2023 during weekdays only. The researcher and or research assistant used patient's registry book to identify those diagnosed with breast cancer where by after their doctor's consultation, participants were asked to willingly participate in the study.

## Variables assessed

Anxiety and depression symptoms among women diagnosed with breast cancer were the main outcome variables. The predictor variables were age, stage of cancer, type of treatment, comorbidities, duration of illness, coping skills, sense of hope, body image, educational level, marital status, social support, and place of residence.

## Data collection method

**Tools and measurements.** We used a questionnaire that consisted of structured questions and tools that were adopted so as to measure the variables. The questionnaire was translated from English to Swahili using Brislin's back-translation method for cross-cultural research. Two bilingual specialists who were conversant in Swahili and English were consulted. To adapt the Swahili questionnaire for cultural differences, we conferred with three mental health and psychiatry specialists (two mental health nurses and one psychiatrist). The questionnaire had section one, two, three, four, and five which entailed structured questions for sociodemographic data, Hospital Anxiety and Depression Scale (HADS), Miltidimensional Scale of Perceived Social Support (MSPSS), State of Hope Scale (SHS), and Emotion Regulation Questionnaire (ERQ), respectively. The structured questionnaire consisted of nine questions which were participant's age, educational level, place of residence, marital status, stage of cancer, comorbidities, comfortability with body image, type of cancer treatment and duration of the illness. To ensure the validity and reliability of the questionnaire used in our study, the questionnaire underwent a thorough review process by experts in the field of psychiatry (two mental health nurses and one psychiatrist) to assess its content validity and ensure that it accurately measures the constructs of interest. Additionally, pilot testing was conducted with a sample of 20 participants to assess the clarity, relevance, and comprehensibility of the questionnaire items. Furthermore, reliability analysis performed to assess the internal consistency of the questionnaire items and ensure that the instrument produces consistent and stable results over time showed Cronbach's alpha 0.72 for Hospital Anxiety Depression Scale – Anxiety (HADS-A), 0.81 for Hospital Anxiety Depression Scale – Depression (HADS-D) and 0.83 for MSPSS. The SHS had internal reliability of Cronbach's α 0.80 for the six-item version questionnaire, 0.82 for the three agency items, and 0.79 for the three pathway items while ERQ had 0.77 and 0.68 for expressive suppression and reappraisal items respectively. By adhering to these rigorous validation procedures, the questionnaire used in our study is deemed to be a valid and reliable tool.

HADS was used to screen for anxiety and depression, it consists of 14 questions, 7 questions for each. Each item scores 0–3, the two variables are scored independently while for both summation of scores to less than 7 are normal levels, 8–10 are mild levels, 11–14 are moderate levels and 15–21 are severe levels of anxiety and or depression. For both variables, individuals with scores above 8 are considered to have anxiety and or depression. It is validated for use in low income countries. Hospital Anxiety Depression Scale – Anxiety (HADS-A) entails questions that screen for anxiety, it gives a specificity of 0.78 and a sensitivity of 0.9 while Hospital Anxiety Depression Scale – Depression (HADS-D) entails questions that screen for depression, it gives a specificity of 0.79 and a sensitivity of 0.83 [14].

MSPSS is a Likert scale with 12 items and 7-response options for each item, and it measures the level of social support. The items are arranged in three subscales which assess sources of social support such as from family members,

friends, or significant others. The total scale score is calculated by summing all item scores and dividing by 12. Perceived social support mean scores are defined as 1–2.9 representing low perceived social support, 3–5 moderate perceived social support, and 5.1–7 high perceived social support [15]. This tool has been validated for use in low-income countries, and showed a Cronbach's α of 0.83 [16].

SHS, a six items tool used to assess sense of hope, has been validated for use in Tanzania [17]. Each item in the tool was scored as 1, 2, 3, 4, 5, 6, 7 and 8 representing definitely false, mostly false, somewhat false, slightly false, slightly true, somewhat true, mostly true and definitely true respectively. The total score ranges from 6–48, with hopefulness increasing as scores increase. The total scores are then transformed into two cutoff points with equal percentiles through visual binning and hence categorized into low, moderate and high states of hope. It has the internal reliability of Cronbach's α 0.82 for the six-item version questionnaire, 0.83 for the three agency items, and 0.74 for the three pathway items [18].

ERQ a 10-item questionnaire assess individual differences and tendencies using two emotion regulation strategies, expressive suppression, and reappraisal. It is rated on a 7-point Likert scale, each item was scored as 1, 2, 3, 4, 5, 6 and 7 representing strongly disagree, somewhat disagree, disagree, neutral, agree somewhat agree and strongly agree. The emotion regulation scores were categorized using the mean score of 28 and 14 for cognitive reappraisal and expressive suppression subsets respectively, therefore those who scored 28 and below were considered as low cognitive reappraisal while those who scored 29 and above were considered as high cognitive reappraisal, as for expressive suppression those who scored 14 and below were considered as low expressive suppression while those who scored 15 and above were considered high expressive suppression. The questionnaire has been validated for use in low and middle income countries and showed good internal consistency with a Cronbach's α of 0.73 and 0.79 for expressive suppression items and reappraisal items respectively [19].

**Data collection procedure.** We used interviewer-administered questionnaire with closed-ended questions to collect data from the study participants. We collected data with the aid of two research assistants who were competent and had prior experience in research activities. The research assistants were trained on study objectives, data collection tools, data collection process, and study's ethical procedures to ensure compliance. The researcher and or research assistant used patients registry book at the nursing station to identify potential participants who met the inclusion criteria and hence they were approached after their doctor's consultation. They were informed about the objective of the study, risks and benefits of their participation in the study, voluntary participation and how long the interview would take. To ensure confidentiality, the women were requested to step in a room that was prepared for interviewing study participants. In the room the women were well informed about the study (by reading the consent form to her) then those who agreed to participate in the study were asked to sign an informed consent form before beginning the interview. The researcher or assistant offered a detailed clarification about matters that were not understood. Each interview used approximately 30–45 minutes. The researcher or research assistant used manual computation to diagnose participants with anxiety and or depression symptoms immediately after completion of filling out the questionnaire. This was done so as to offer counselling and or referral services to those with moderate and or severe levels of anxiety and or depression symptoms.

## Data analysis

Data were entered, cleaned, coded and processed using Statistical Package for Social Sciences (IBM SPSS) version 23. Descriptive statistics were summarized using mean and standard deviation for continuous variables while percentages and frequency were used for categorical variables.

The outcomes of interest anxiety and depression symptoms were measured using HADS tool. In order to estimate and analize the prevalence of anxiety and depression symptoms, scores from the tools were dichotomized into either presence or absence of the symptoms. In this index study a cut-off point of equal to and above 8 was used, where the participants who scored seven and below on either of the subscales was defined as having no symptoms while those who scored 8 and above was defines as having the symptoms [20,21].

The socio-demographic variables, age was grouped into four categories: youths (15–24), young adults (25–44), middle aged adults (45–60), elderly (61 and above) [22]. This age categorization has been used in different researches involving women diagnosed with breast cancer and is also in-keeping with WHO categories [22]. Education level was categorized into primary education level and below and ordinary education level and above [7]. Participants marital status was also dichotomized into two: without partner (single/divorced/separated/widow) and with partner (cohabiting/married) [6]. The place of residence was categorized into residing within the city or outside the city. The participants stage of cancer was categorized into I do not know, stage 1&2 and stage 3&4. Comorbidities was categorized into no, I don't have and yes I do have, while comfortability with body image was categorized into no, I don't and yes I do. The type of cancer treatment one is receiving was categorized into chemotherapy, radiotherapy and pills and supplements. Participants duration since diagnosis was categorized into below three years and above three years. Perceived social support was dichotomized into 1–5 as low perceived social support and 5.1–7 as high perceived social support [23]. For the state of hope, after obtaining their total scores it was then transformed into two cutoff points with equal percentiles through visual binning (which were below 33.33% and above 66.66%), hence categorizing into low, moderate and high states of hope [24]. The emotion regulation scores were categorized using the mean score of 28 and 14 for cognitive reappraisal and expressive suppression subsets respectively, therefore those who scored 28 and below were considered as low cognitive reappraisal while those who scored 29 and above were considered as high cognitive reappraisal, as for expressive suppression those who scored 14 and below were considered as low expressive suppression while those who scored 15 and above were considered high expressive suppression [25]. Bivariate analysis was performed to determine the association between biopsychosocial characteristics and anxiety and depression symptoms. Moreover, multivariable binary logistic regression analysis to determine the predictors of anxiety and depression symptoms. In all analyses, a *p*-value of <0.05 was considered statistically significant.

### Ethical issues and research clearance

We obtained ethical clearance to conduct this study from Muhimbili University of Health and Allied Sciences – Institution Review Board (MUHAS – IRB) with reference number MUHAS-REC-04-2023-1622. Permission was obtained from the head of research and training at ORCI. Potential participants were well informed about the study, and those who provided signed consent forms were recruited. Voluntary participation was emphasized and participants were informed that there would be no consequences on refusal to participate and/or withdrawal of consent at any time. Confidentiality was established through anonymity; therefore, questionnaires included codes to protect their identity rather than names. Participants who screened positive for depression and or anxiety with moderate and severe levels were reported to the nurse in charge for further management while those with mild levels were engaged in a brief talk therapy with the researcher to manage symptoms earlier.

## Results

The Sociodemographic(s), biological, and psychological characteristics of the study participants.

A total of 384 study participants had their questionnaires filled. Their mean (SD) age was 49.66 (11.69) years. The majority 179 (46.6%) were aged 45–60, and 273 (71.1%) had primary educational level and below. More than half (229; 59.6%) had no partners and 230 (59.9%) resided outside the city of Dar-es-Salaam. Only 174 (45.3%) had high perceived social support. The majority of participants 312 (81.3%), 308 (80.2%) and 292 (76%) were receiving chemotherapy, below three years of breast cancer diagnosis and didn't have comorbidities, respectively. Half 214 (55.7%), 136 (35.4%) and 199 (51.5%) were comfortable with their body image, with low state of hope and low expressive suppression respectively "Table 1".

### Anxiety and depression symptoms of the study participants

The prevalence of anxiety and depression symptoms was 44.8% and 50.5%, respectively. Among the participants who had anxiety 31.5% and 13.3% had moderate and severe levels of anxiety symptoms respectively. Among the participants with depression 38% and 12.5% had moderate and severe levels of depression symptoms respectively "Table 2".

**Table 1. Frequency distribution of study participants' baseline characteristics (N = 384).**

| Variable | Frequency (n) | Percentage (%) |
|---|---|---|
| Age (years) | | |
| 25-44 | 138 | 35.9 |
| 45-60 | **179** | **46.6** |
| 61 and above | 67 | 17.4 |
| Education level | | |
| Primary education level and below | **273** | **71.1** |
| Ordinary level and above | 111 | 28.9 |
| Marital status | | |
| Without partner | 155 | 40.4 |
| With partner | **229** | **59.6** |
| Residence | | |
| Outside the city | **230** | **59.9** |
| Inside the city | 154 | 40.1 |
| Stage of cancer | | |
| I don't know | **237** | **61.7** |
| Stage 1 and 2 | 104 | 26.1 |
| Stage 3 and 4 | 43 | 11.2 |
| Comorbidities | | |
| No | **292** | **76** |
| Yes | 92 | 24 |
| Comfort ability with body image | | |
| No | 170 | 44.3 |
| Yes | **214** | **55.7** |
| Type of Cancer Treatment | | |
| Chemotherapy | **312** | **81.3** |
| Radiotherapy | 35 | 9.1 |
| Pills and supplements | 37 | 9.6 |
| Duration of illness | | |
| Below 3 years | **308** | **80.2** |
| Above 3 years | 76 | 19.8 |
| Perceived social support | | |
| Low support | 42 | 10.9 |
| Moderate Support | 168 | 43.8 |
| High support | **174** | **45.3** |
| State of hope | | |
| Low | **136** | **35.4** |
| Moderate | 127 | 33.1 |
| High | 121 | 31.5 |
| Emotion regulation | | |
| *Cognitive reappraisal* | | |
| Low | **194** | **50.5** |
| High | 190 | 49.5 |
| *Expressive suppression* | | |
| Low | **199** | **51.5** |
| High | 185 | 48.2 |

**Table 2. Anxiety and depression symptoms (N = 384).**

| Variable | Frequency (n) | Percentage (%) |
|---|---|---|
| Anxiety symptoms | | |
| Yes | **172** | **44.8** |
| No | 212 | 55.2 |
| Anxiety symptoms severity | | |
| Mild | 212 | 55.2 |
| Moderate | **121** | **31.5** |
| Severe | **51** | **13.3** |
| Depression symptoms | | |
| Yes | **194** | **50.5** |
| No | 190 | 49.5 |
| Depression symptoms severity | | |
| Mild | 190 | 49.5 |
| Moderate | **146** | **38** |
| Severe | **48** | **12.5** |

## Factors associated with anxiety and depression symptoms

**Bivariate analysis of factors associated with anxiety and depression symptoms among women diagnosed with breast cancer.** Those without a partner had approximately two times higher odds of developing anxiety and depression symptoms [crude Odds Ratio (cOR): 1.59; 95%Confidence Interval (CI): 1.05, 2.40, p = 0.027] and [cOR: 2.49; 95%CI: 1.64, 3.80; p < 0.001] respectively as compared to those who had partners. Participants who did not know the stage of cancer had lesser odds of having anxiety and depression symptoms [cOR: 0.40; 95%CI: 0.20, 0.78; p = 0.008] and [cOR: 0.38; 95%CI: 0.19, 0.80; p = 0.010] respectively while those in stage one and two [cOR: 0.19; 95%CI: 0.86, 0.42; p < 0.001] showed lesser odds of developing depression symptoms. Both low and moderate states of hope were found to have lesser odds of having anxiety symptoms [cOR: 0.57; 95%CI: 0.35, 0.93; p = 0.024] and [cOR: 0.56; 95%CI: 0.34, 0.93; p = 0.024] respectively as compared to those with high state of hope. Those with low cognitive reappraisal showed approximately two times higher odds of having anxiety and depression symptoms [cOR: 1.82; 95%CI: 1.21, 2.74; p = 0.004] and [cOR: 1.65; 95%CI: 1.10, 2.47; p = 0.015] respectively as compared to those with high cognitive reappraisal. Those receiving radiotherapy showed approximately four times higher odds of developing depression symptoms [cOR: 3.94; 95%CI: 1.20, 8.25; p = 0.020] as compared to those receiving chemotherapy and pills and supplements.

**Multivariable analysis of factors associated with and anxiety and depression symptoms among women diagnosed with breast cancer.** Not knowing the stage of cancer had lesser odds of having anxiety and depression symptoms [adjusted Odds Ratio (aOR): 0.39; 95%CI: 0.19, 0.83; p = 0.014] and [aOR: 0.38; 95%CI: 0.17, 0.86; p = 0.020] respectively while those in stage one and two [aOR: 0.36; 95%CI: 0.16, 0.81; p = 0.014] had lesser odds of having anxiety symptoms as compared to those who had cancer stage 3 & 4. Respondents who had low levels of hope were found to have lesser odds of developing anxiety symptoms [aOR: 0.52; 95%CI: 0.31, 0.86; p = 0.012] as compare to those who had moderate and high states of hope. Emotion regulation of the subset cognitive reappraisal from the group of low cognitive reappraisal showed two times higher odds of having anxiety and depression symptoms [aOR: 2.09; 95%CI: 1.33, 3.28; p < 0.001] and [aOR: 2.41; 95%CI: 1.52, 3.83; p < 0.001] as compared to those with high cognitive reappraisal. Those receiving radiotherapy type of cancer treatment showed approximately four times higher odds of developing depression symptoms [aOR: 3.86; 95% CI: 1.34, 11.08; p = 0.012] as compared to those receiving other types of treatment modalities while those without partners showed approximately three times higher odds of developing depression symptoms [aOR: 2.56; 95%CI: 1.59, 4.10; p < 0.001] as compared to those with partners. The bivariate and multivariable logistic regression analyses are presented in "Table 3" below:

**Table 3. Bivariate and Multivariable analysis of factors associated with anxiety and depression symptoms among women diagnosed with breast cancer (N = 384).**

| Variable | Anxiety symptoms | | | | | Depression symptoms | | | | |
|---|---|---|---|---|---|---|---|---|---|---|
| | n(%) | Bivariate analysis | | Multivariable analysis | | n(%) | Bivariate analysis | | Multivariable analysis | |
| | | cOR (95%CI) | p-value | aOR (95%CI) | p-value | | cOR (95%CI) | p-value | aOR (95%CI) | p-value |
| **Age (years)** | | | | | | | | | | |
| 25–44 | 71 (40.28) | 1.09 (0.61–1.06) | 0.768 | 1.41 (0.75 - 2.66) | 0.292 | 73 (37.63) | 0.81 (0.45 - 1.45) | 0.474 | 0.82 (0.42 - 1.59) | 0.556 |
| 45–60 | 68 (39.53) | 0.63 (0.36–1.11) | 0.111 | 0.85 (0.46 - 1.58) | 0.611 | 82 (42.27) | 0.61 (0.34 - 1.07) | 0.85 | 0.84 (0.45 - 1.59) | 0.597 |
| 61 and above | 33 (19.18) | Ref | | Ref | | 39 (20.10 | Ref | | Ref | |
| **Education level** | | | | | | | | | | |
| Primary education level and below | 122 (70.93) | 0.99 (0.63–1.54) | 0.95 | 1.34 (0.81 - 2.23) | 0.265 | 141 (72.68) | 1.17 (0.75–1.82) | 0.488 | | |
| Ordinary level and above | 50 (29.07) | Ref | | Ref | | 53 (27.32) | Ref | | | |
| **Marital status** | | | | | | | | | | |
| Without partner | 80 (46.51) | 1.59 (1.05–2.40) | **0.027*** | 1.55 (0.98–2.45) | 0.060 | 99 (51.03) | 2.49 (1.64–3.80) | **0.000*** | 2.56 (1.59 - 4.10) | 0.000*** |
| With partner | 92 (53.49) | Ref | | Ref | | 95 (48.97) | Ref | | Ref | |
| **Residence** | | | | | | | | | | |
| Outside the city | 100 (58.14) | 0.88 (0.58–1.32) | 0.527 | | | 110 (56.70) | 0.76 (0.51–1.15) | 0.197 | 0.73 (0.46 - 1.16) | 0.186 |
| Inside the city | 72 (41.86) | Ref | | | | 84 (43.30) | Ref | | Ref | |
| **Stage of cancer** | | | | | | | | | | |
| I don't know | 101 (58.72) | 0.40 (0.20–0.78) | **0.008**** | 0.39 (0.19–0.83) | **0.014**** | 125 (64.43) | 0.38 (0.19–0.80) | **0.010*** | 0.38 (0.17 - 0.86) | 0.020* |
| Stage 1 and 2 | 43 (25.00) | 0.38 (0.18–0.79) | 0.10 | 0.36 (0.16–0.81) | **0.014**** | 37 (19.07) | 0.19 (0.86 - 0.42) | **0.000*** | 0.16 (0.07 - 0.38) | 0.158 |
| Stage 3 and 4 | 28 (16.28) | Ref | | Ref | | 32 (16.50) | Ref | | Ref | |
| **Comorbidities** | | | | | | | | | | |
| No | 137 (79.65) | Ref | | | | 149 (76.80) | Ref | | | |
| Yes | 35 (20.35) | 0.70 (0.43–1.12) | 0.695 | | | 45 (23.20) | 0.92 (0.58–1.47) | 0.724 | | |
| **Comfort ability with body image** | | | | | | | | | | |
| No | 77 (44.77) | 1.037 (0.69–1.56) | 0.860 | 1.14 (0.72 - 1.81) | 0.582 | 81 (41.75) | 0.81 (0.54–1.22) | 0.316 | | |
| Yes | 95 (55.23) | Ref | | Ref | | 113 (58.25) | Ref | | | |
| **Type of cancer treatment** | | | | | | | | | | |
| Chemotherapy | 146 (84.88) | 2.38 (1.11–5.07) | 0.25 | 1.68 (0.73 - 3.86) | 0.222 | 157 (80.93) | 1.66 (0.83 - 3.35) | 0.154 | 1.29 (0.59 - 2.84) | 0.530 |
| Radiotherapy | 16 (9.30) | 2.27 (0.85–6.08) | 0.102 | 1.954 (0.68 - 5.63) | 0.215 | 23 (11.86) | 3.149 (1.20 - 8.25) | **0.020*** | 3.86 (1.34–11.08) | 0.012** |
| Pills and supplements | 10 (5.82) | Ref | | Ref | | 14 (7.21) | Ref | | Ref | |
| **Duration of illness** | | | | | | | | | | |
| Below 3 years | 138 (80.23) | Ref | | Ref | | 161 (82.99) | Ref | | Ref | |
| Above 3 years | 34 (19.77) | 1.00 (0.60–1.65) | 0.991 | 1.09 (0.62 - 1.91) | 0.765 | 33 (17.01) | 0.70 (0.42 - 1.16) | 0.168 | 0.62 (0.35 - 1.11) | 0.107 |

*(Continued)*

**Table 3.** (Continued)

| Variable | Anxiety symptoms | | | | | Depression symptoms | | | | |
|---|---|---|---|---|---|---|---|---|---|---|
| | n(%) | Bivariate analysis | | Multivariable analysis | | n(%) | Bivariate analysis | | Multivariable analysis | |
| | | cOR (95%CI) | p-value | aOR (95%CI) | p-value | | cOR (95%CI) | p-value | aOR (95%CI) | p-value |
| **Perceived social support** | | | | | | | | | | |
| Low support | 85 (49.42) | 0.71 (0.47 - 1.06) | 0.093 | 0.71 (0.44 - 1.15) | 0.166 | 104 (53.61) | 0.96 (0.64 - 1.43) | 0.824 | 0.83 (0.51 - 1.35) | 0.556 |
| High support | 87 (50.58) | Ref | | Ref | | 90 (46.39) | Ref | | Ref | |
| **State of hope** | | | | | | | | | | |
| Low | 55 (31.78) | 0.57 (0.35–0.93) | **0.024*** | 0.50 (0.78–0.90) | **0.021*** | 62 (31.96) | 0.75 (0.46 - 1.22) | 0.243 | 0.70 (0.38 - 1.28) | 0.244 |
| Moderate | 51 (29.65) | 0.56 (0.34–0.93) | **0.024*** | 0.58 (0.33–1.02) | 0.056 | 68 (35.05) | 1.03 (0.62 - 1.69) | 0.918 | 1.258 (0.71 - 2.23) | 0.432 |
| High | 66 (38.37) | Ref | | Ref | | 64 (32.99) | Ref | | Ref | |
| **Emotion regulation** | | | | | | | | | | |
| Cognitive reappraisal | | | | | | | | | | |
| Low | 101 (58.72) | 1.82 (1.21–2.74) | **0.004*** | 2.09 (1.33–3.28) | **0.001*** | 110 (56.70) | 1.65 (1.10 - 2.47) | **0.015*** | 2.41 (1.52 - 3.83) | **0.000*** |
| High | 71 (41.28) | Ref | | Ref | | 84 (43.30) | Ref | | Ref | |
| Expressive suppression | | | | | | | | | | |
| Low | 88 (51.16) | Ref | | | | | | | Ref | |
| High | 84 (48.84) | 1.05 (0.70–1.60) | 0.816 | | | | | | | |

Key: CI = Confidence Interval, cOR= crude Odds Ratio, aOR= adjusted Odds Ratio, ρ=p-value, *=ρ<0.05, **=ρ<0.01, ***=ρ<0.001.

## Discussion

This study aimed at determining the prevalence and factors associated with anxiety and depression symptoms among women diagnosed with breast cancer at the ORCI, Dar es Salaam. In our study, both prevalences of anxiety (44.8%) and depression (50.5%) were found to be relatively high in women diagnosed with breast cancer. Participants who didn't know the stage of their cancer, in early stage of cancer (1&2), with low state of hope and low cognitive reappraisal were significantly associated with anxiety symptoms. Those without partners, did not know stage of their cancer, received radiotherapy treatment and had a low cognitive reappraisal, were significantly associated with depression symptoms. Hence, screening for anxiety and depression symptoms and incorporating proper psychological or psychiatric treatment for women diagnosed with breast cancer is crucial.

The prevalence of anxiety and depression symptoms in the general population globally was 2.9%–5.8% and 3.6%–5.4%, respectively [26]. In our study, the prevalence of anxiety and depression were significantly high among women diagnosed with breast cancer. This highlight the psychological pain that comes with receiving a cancer diagnosis even in the face of substantial advancements in treatment options. Similar findings were observed in France where the prevalence of women with breast cancer diagnosed with anxiety and depression were 43.4% and 56.2% respectively [27]. The similarity could be explained by the use of similar methodologies, diagnostic tools and cutoff points. In contrary results from Asia and USA indicates a lower prevalence of anxiety and depression symptoms with 32.2% in Asia, 4.1% in the USA and 38.2% in Asia, 31% − 37% in the USA, respectively [2,4,11,28]. Moreover, the different cultural backgrounds, a vast of health facilities, advanced technologies used in different tests, treatment modalities and health-seeking behavior

in the developed countries as compared to the low-income countries could explain the differences [26,29]. This calls for consolidation of oncological and mental health services as a crucial step in helping women diagnosed with breast cancer.

According to our study, women with breast cancer who had cognitive reappraisal had nearly doubled the odds to have anxiety and depression symptoms. This finding suggests that these women might be more vulnerable to increase anxiety and depression if they reinterpret stressful experience negatively. Similar are the findings of Benson et al, who did a study in Ghana using brief cope tests to investigate coping strategies [1]. Furthermore the odds of anxiety and depression symptoms is increased by elements like hopelessness, a lack of social support from friends, family, or significant others, and stigmatization [30]. These results point to the urgent need for focused therapies that strengthen cognitive appraisal techniques in this susceptible group.

Being without a partner was found to approximately triples the odds of developing depression symptoms as compared to those with partners. In our opinion, besides lacking a partner, altered body image after surgery which trigger feelings of low self-esteem, the worry of being unable to get a spouse, have children and breast feed again, uncertainty about their future and health may have contributed to their psychiatric morbidity. Alike are studies done in France and Ghana which found that being with a partner is a protective factor against developing anxiety and depression symptoms among women diagnosed with breast cancer [5,30]. Regardless of one's cultural background, these similarities may be explained by the necessity and significance of having a significant other to rely on and share sorrows with during difficult times. The risk is further advanced by other factors like a lack of social support, financial limitations, and a fear of dying before attaining goals desired [31].

Study participants who reported receiving radiotherapy, type of cancer treatment were had approximately quadrupled odds of developing depression. The repercussion of these findings confirm that not only does cancer itself have a significant impact on the women's mental condition and general quality of life, but the available treatment alternatives as well. These results were in harmony with research from Australia, Spain, and London that found some cancer treatment therapies are more likely to raise the risk of depression [9,32,33]. On the other hand, compared to other treatment modalities, these three trials showed that chemotherapy was associated with higher incidence of depression. The extended exposure to the medical environment due to daily sessions of radiotherapy over a few weeks as compared to once a week in chemotherapy may account for the variation in cancer treatment types that are likely to raise the odds for developing depression [3,31–33]. Educating and informing the patient about every step of the treatment process and expected outcomes should be an act of high priority, this will help lessen the feelings of uncertainty and hopelessness.

Individuals who were uncertain about their cancer stage exhibited statistically significant associations with symptoms of anxiety and depression, suggesting that this uncertainty may serve as a protective factor. This experience shows that patients tend to become less preoccupied with prognostic factors or the potential outcomes linked to various stages of their illness. Supporting this notion, a study by Sahu et al. found that patients in treatment sensed a notable decrease in symptoms of depression and anxiety. This observation may explain why these patients demonstrated a decreased worry regarding their cancer stage and its implications; their focus shifted to the treatment process and hope for recovery rather than the uncertainties of their diagnosis [34].

Participants in the early stage of breast cancer (stage 1&2) were less likely to have a risk of developing anxiety symptoms. Early stage being a protective factor suggests that patients may have a more positive attitude and hope for good prognosis in the early stages of breast cancer. This is contrary to two studies done in Asia and Saudi Arabia which found having an early stage of cancer is rather a risk to developing anxiety symptoms than a protective factor [2,35]. However, the sudden revelation of a serious health condition can trigger feelings of fear, uncertainty, and anxiety about the future, regardless of the stage of cancer [28,34]. This calls for an increased promotion of awareness about breast cancer and early screening.

Surprisingly individuals with low levels of hope had lesser odds of experiencing symptoms of anxiety. The feeling of helplessness, despair, lack of enthusiasm for the future, and detachment of emotions, which results in emotional

numbness, are characteristics of a low level of hope. This lessens the strength of emotional reactions and, as a result, the likelihood of experiencing anxiety symptoms, emotional numbness has been proven to accommodate people in managing negative emotions [19]. Despite its effectiveness in preventing the patients from experiencing heightened levels of anxiety, it is still a bad coping mechanism and can be more harmful to the mental health of the patient in the long run [1]. Therefore, incorporating psychological therapies to help these patients emotionally is crucial.

The study used a cross-sectional design, in which the direction of causality could not be determined. However, the study gave the prevalence and associated factors with anxiety and depression symptoms among women diagnosed with breast cancer. The use of face-to-face interviews which were conducted by the researcher and research assistants during data collection, could have developed information bias through socially desirable answers. Participants could have under- or over-reported for example, responses about perceived social support due to the expectation of gaining private support after the study. To overcome this, participants were clearly informed about the significance of the study so as to enhance their sincerity.

## Conclusion

The findings reveal a complex interplay among psychological resilience, protocols of treatment, and social support in the mental health of women diagnosed breast cancer. The protective factors that have been identified, such as early diagnosis, underlines how necessary it is to root for positivity and provide patients with enough information about the stages of cancer disease. However, targeted interventions are very crucial due to the high vulnerability of patients going through radiation therapy as their mode of treatment and lack social support. Given the heightened prevalence of anxiety and depression in women diagnosed with breast cancer, it is imperative that healthcare providers, especially physicians and nurses, routinely screen women with breast cancer for these mental health issues. The development of novel therapeutic strategies that address the psychological difficulties these women experience during their cancer treatment and journey in general is just as important as the ongoing developments in chemo therapy, hormone therapy, and radiation therapy. The consolidation of mental health services into current cancer care frameworks should be given top priority by cancer centres, which should employ mental health nurses, psychologists, and psychiatrists to provide specialised care that women diagnosed with breast cancer need. The anxiety and depression symptoms experienced by women with breast cancer can be considerably reduced with this interdisciplinary approach. Moreover, this study could serve as a foundational reference for future implementation research, including case-control longitudinal studies, that will aim at evaluating the effect of incorporating targeted psychological therapies in the cancer care framework. Such studies could provide valuable insights into effective strategies for supporting women diagnosed with breast cancer who are struggling with anxiety and depression, and ultimately enhancing their overall quality of life.

## Acknowledgments

We would like to thank all the participants for allowing us to conduct the interview. We also acknowledge the research assistants involved in the data collection process. Finally, we thank the Ocean Road Cancer Institute administration for the permission to conduct this study.

## Author contributions

**Conceptualization:** Ummy Msenga.

**Formal analysis:** Ummy Msenga.

**Methodology:** Ummy Msenga.

**Supervision:** Joel S. Ambikile, Salome E. Buluba.

**Validation:** Joel S. Ambikile, Salome E. Buluba.

**Writing – original draft:** Ummy Msenga.

**Writing – review & editing:** Ummy Msenga.

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
