## [Decision Letter · Decision Letter 0]

Dear Dr. Msenga,

Thank you for submitting your manuscript to PLOS ONE. After careful consideration, we feel that it has merit but does not fully meet PLOS ONE’s publication criteria as it currently stands. Therefore, we invite you to submit a revised version of the manuscript that addresses the points raised during the review process.

The introduction adequately sets the stage by presenting the significance of anxiety and depression in breast cancer patients, particularly in low-income countries. However, it could strengthen the rationale by expanding on the gap in local Tanzanian research beyond referencing socioeconomic factors. A clearer link between this gap and the current study would add value.

It would be beneficial to compare the prevalence rates found in this study with global or regional studies more thoroughly in the results section. Although global prevalence is briefly mentioned in the introduction, it would provide a more direct assessment of whether the rates observed in Tanzania are higher or lower than expected.

The conclusion appropriately emphasizes the need for mental health screening and intervention, but it could benefit from more specific recommendations on integrating these services into existing cancer care frameworks in Tanzania.

We look forward to receiving your revised manuscript.

Kind regards,

Mikiyas Amare Getu

Academic Editor

PLOS ONE

Journal Requirements:

3. When completing the data availability statement of the submission form, you indicated that you will make your data available on acceptance. We strongly recommend all authors decide on a data sharing plan before acceptance, as the process can be lengthy and hold up publication timelines. Please note that, though access restrictions are acceptable now, your entire data will need to be made freely accessible if your manuscript is accepted for publication. This policy applies to all data except where public deposition would breach compliance with the protocol approved by your research ethics board. If you are unable to adhere to our open data policy, please kindly revise your statement to explain your reasoning and we will seek the editor's input on an exemption. Please be assured that, once you have provided your new statement, the assessment of your exemption will not hold up the peer review

Additional Editor Comments (if provided):

The introduction adequately sets the stage by presenting the significance of anxiety and depression in breast cancer patients, particularly in low-income countries. However, it could strengthen the rationale by expanding on the gap in local Tanzanian research beyond referencing socioeconomic factors. A clearer link between this gap and the current study would add value.

It would be beneficial to compare the prevalence rates found in this study with global or regional studies more thoroughly in the results section. Although global prevalence is briefly mentioned in the introduction, it would provide a more direct assessment of whether the rates observed in Tanzania are higher or lower than expected.

The conclusion appropriately emphasizes the need for mental health screening and intervention, but it could benefit from more specific recommendations on integrating these services into existing cancer care frameworks in Tanzania.

Reviewers' comments:

Reviewer's Responses to Questions

**Comments to the Author**

1. Is the manuscript technically sound, and do the data support the conclusions?

Reviewer #1: Yes

Reviewer #2: Yes

Reviewer #3: No

2. Has the statistical analysis been performed appropriately and rigorously?

Reviewer #1: No

Reviewer #2: Yes

Reviewer #3: No

3. Have the authors made all data underlying the findings in their manuscript fully available?

Reviewer #1: Yes

Reviewer #2: No

Reviewer #3: No

4. Is the manuscript presented in an intelligible fashion and written in standard English?

Reviewer #1: No

Reviewer #2: Yes

Reviewer #3: No

Reviewer #1: Dear Editor, thank you very much for inviting me to review this important topic that aligns with my specialty. I also appreciate authors for bringing such an interesting and current global concern topic.

Even though the manuscript has merit, it needs a major revision, especially on the method and result sections.

1. In the abstract section, please coordinate those mental health problems with breast cancer to attract the reader...

2. Your data collection period is between May and 17 June 2023; please specify the date.

3. Your recommendation is poor. Please rewrite your recommendation based on your findings.

4. In sample size calculation, why you didn’t add non-response rate...

5. Please clearly put your operational definitions...

6. How do you select your study participants? Who diagnosed your study participants???

7. In the result section, your response rate was 100%... Are you sure???

8. It’s not clear how you categorize the variables age, marital status, and educational status.

9. Who reports the stage of cancer????

10. In binary tables, how do you select the variables for multivariable? For instance, age is not significantly associated with the crude odds ratio, so how is it a candidate for AOR???

11. Please revise BINARY TABLE...

12. Discussion is poor.

13. Rewrite your conclusion and recommendations... It needs revision. Your recommendation should align with your findings.

Reviewer #2: I do not have major concern with the technical content of the paper. However, there are some concerns

a) The introductioon section did not provide convincing justification on the need for this study. Given there are already available evidences on prevamence of anxiety and depression in conneciton to breast cancer, why should this study be of interest? This is not clear from the introduction

2) The findings from this study presented factors associated with and anxiety and depression. However, similar findings are out there in Sub-Saharan Countries. Again these findings are not any different from what is availble. Again this is critical limitaiton of this study

3) the conclusive argument on implicaion of anxiety and depression doesn't seem to offer any useful advice to researchers, policy makers and programmers. So, this study doesn't have novelity.

Reviewer #3: GENERAL COMMENTS/ QUERIES/ RECOMMENDATIONS:

1- A good and important topic for clinical and public health domains.

2- A vote of thanks to authors for considering "patients' perspectives" in their clinical research findings. A commendable attitude!

3. However, there are notable errors/questionable statements/sentences that are at best contradictory and at worst fallacious in composition throughout the manuscript sections!

e.g. in results section, what has been reported as “bivariate analyses” are in actual fact UNIVARIATE/CRUDE ANALYSES and what went on as “multivariate analyses” in actual fact were MULTIVARIABLE ANALYSES!

Besides, findings of univariate & multivariable binary logistic regression suggest POTENTIAL BUT SIGNIFICANT confounding!!!

Otherwise, it was not clear (and some justifications FALLACIOUS!) on how/which criteria were used to dichotomise variables during analysis. For instance, it is not true that UN considers elderly people as >61 years old (fact: people >65 years are considered 'elderly' under current UN definition)

Recommendations: MINOR correction but consider (in addition to specific comments attached separately to editors):

1. to re-write the manuscript (see specific comments/queries/recommendations attached to editors!) once again before it could MERIT consideration for publication at PLOS ONE journal.

2. Authors should attach RAW DATASET (preferably as an APPENDIX) for verification of their analyses in lieu of the current picture displayed in the results section of the manuscript!

**Do you want your identity to be public for this peer review?** For information about this choice, including consent withdrawal, please see our Privacy Policy

Reviewer #1: **Yes: ** Solomon Seyife Alemu

Reviewer #2: **Yes: ** Mirgissa Kaba

Reviewer #3: **Yes: ** Kelvin Melkizedeck Leshabari

---

## [Author Response · Author response to Decision Letter 1]

12 May 2025

Response to reviewers is attached with this revision

---

## [Editor Report · Decision Letter 1]

Prevalence of anxiety and depression symptoms and associated factors among women diagnosed with breast cancer: A cross-sectional study at Ocean Road Cancer Institute in Dar-es-Salaam - Tanzania.

PONE-D-24-37916R1

Dear Dr. Msenga,

We’re pleased to inform you that your manuscript has been judged scientifically suitable for publication and will be formally accepted for publication once it meets all outstanding technical requirements.

Kind regards,

Mikiyas Amare Getu

Academic Editor

PLOS ONE
---

## [Editor Report · Acceptance letter]

PONE-D-24-37916R1

PLOS ONE

Dear Dr. Msenga,

I'm pleased to inform you that your manuscript has been deemed suitable for publication in PLOS ONE. Congratulations! Your manuscript is now being handed over to our production team.

Kind regards,

on behalf of

Dr. Mikiyas Amare Getu

Academic Editor

PLOS ONE